# Rumor Detection Based on SAGNN: Simplified Aggregation Graph Neural Networks

**Liang Zhang** [1] , **Jingqun Li** [2,*], **Bin Zhou** [1] and **Yan Jia** [1]

1 College of Computer, National University of Defense Technology, Changsha 410073, China; gfkdzliang@163.com (L.Z.); binzhou@nudt.edu.cn (B.Z.); jiayan@nudt.edu.cn (Y.J.)
2 Shenzhen LiCi Electronic Company, Shenzhen 518000, China
* Correspondence: lij213@mcmaster.ca

**Abstract:** Identifying fake news on media has been an important issue. This is especially true considering the wide spread of rumors on popular social networks such as Twitter. Various kinds of techniques have been proposed for automatic rumor detection. In this work, we study the application of graph neural networks for rumor classification at a lower level, instead of applying existing neural network architectures to detect rumors. The responses to true rumors and false rumors display distinct characteristics. This suggests that it is essential to capture such interactions in an effective manner for a deep learning network to achieve better rumor detection performance. To this end we present a simplified aggregation graph neural network architecture. Experiments on publicly available Twitter datasets demonstrate that the proposed network has performance on a par with or even better than that of state-of-the-art graph convolutional networks, while significantly reducing the computational complexity.

**Keywords:** rumor detection; graph neural network; artificial intelligence

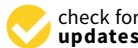

## 1. Introduction

A rumor involves the communication of information that has not been confirmed by a reliable source [1]. While true rumors tell truth, false rumors are those which communicate fabricated news, see Figure 1. Identifying false rumors has now become a major concern for effective use of social media like Twitter and Instagram because of the popularity and easy accessibility of these social media platforms. Rumors can propagate very fast and might have big negative impacts on the society. However, it is a complicated matter to identify rumours from massive amounts of online information. Therefore, it is necessary and highly desirable to develop automatic approaches in order to detect rumors at an early stage so as to mitigate their damages.

As suggested by Figure 1, different class of rumors and their responses usually display different characteristics [2]. As a consequence, it is possible to extract features of some sort from the posts and their responses for the purpose of classifying rumors. In fact, early studies on automatic rumor detection mainly focused on designing effective features from various information sources, including text content, publisher's profiles, and propagation patterns [3,4]. However, these feature-based methods are extremely time-consuming, biased, and labor-intensive. Furthermore, they are lack of robustness: if one or several types of hand-crafted features are unavailable, inadequate or manipulated, the effectiveness of these approaches will be affected.

Motivated by the success of deep learning, many recent studies apply various neural networks for rumor detection. For example, recurrent neural network [5] is applied to learn a representation of tweet text over the post time. The latest efforts focus on applying graph learning techniques for rumor detection [6,7], due to the rapid development of graph neural networks in recent years and the fact that posts on social media are naturally structured as graphs.

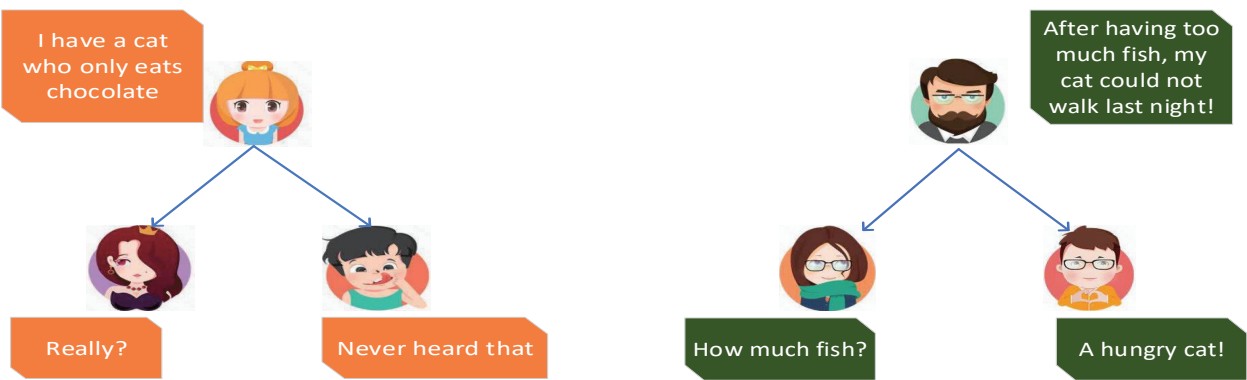

(**a**) A false rumor: the source post is actually false    (**b**) A true rumor: the source post is actually true

**Figure 1.** Rumors and example responses.

In this work, we propose an efficient method of detecting rumors. The contribution of the paper is as follows.

First, we propose a graph neural network architecture to capture the interaction through a trainable aggregation layer. The efficiency and effectiveness are achieved through a reduced convolution layer, called the aggregation layer in the rest of the paper. While a conventional convolutional layer contains $N^2$ learnable parameters, where $N$ is the dimension of the embedded features, the simplified aggregation layer has only two parameters. Since $N$ is usually something of the order of 100, we see that the proposed aggregation layer significantly reduces the computation complexity of the training process, especially considering the calculation of the gradients of the parameters of the embedding layer, which is usually the first layer of the network, using the back propagation algorithm.

Second, we demonstrate that the performance of the proposed simplified network architecture is on a par with or even better than the more complicated conventional graph convolutional network.

The rest of the paper is organized in the following way: First, we review the related rumor detection algorithms in Section 2. Then the proposed simple aggregation network architecture is presented, along with the learning procedure. Next, we give experiment results of applying the SAGNN architecture to two publicly available Twitter datasets. Finally, some conclusions are drawn about the simple aggregation network and its possible applications

## 2. Related Work

Early rumor detection studies were based on hand-crafted features extracted from text content, users' profile and temporal information. For example, features based on the text contents, users, topics and propagation patterns of messages were used to measure the credibility of news on Twitter [2]. Temporal characteristics of the features were explored in [3] to incorporate various social context information, based on the time series of rumor's life cycle. Another development was the propagation trees related methods which focus on the differences in the characteristics of real and false information transmission. These include the kernel-based method [4] in which a propagation tree kernel was proposed to capture high-order patterns differentiating different types of rumors by evaluating the similarities between their propagation tree structures.

The performance of the algorithms based on these hand-crafted features or propagation characteristics are relatively low and sensible to noises. Also, it is labour intensive and very time consuming to devise effective features by hand. See [4,8] for a more detailed account of feature-based and propagation tree related methods.

To address the above difficulties, deep learning models are studied in recent years to learn efficient features for classifying rumors in an automatic manner. Ma et al. [5] first presented a recurrent neural networks (RNN) based model to learn text representations of relevant posts over time. Later on, a recursive network architecture was proposed in [9].

In their latest work [10], a GAN (Generative Adversarial Networks) based architecture was proposed.

A parallel development is the CNN (Convolutional Neural Network) based methods. Yu et al. [11] proposed a convolutional method for misinformation identification based on CNNs , which can capture high-level interactions among significant features. A multi-module model was proposed in [12] to capture the text and user characteristics of messages. Liu et al. modelled the propagation path as multivariate time series, and applied both recurrent and convolutional networks to capture the variations of user characteristics along the propagation path [13].

A comparison study of deep learning rumor detection algorithms was conducted in [14], in which the performances of ten different deep learning architectures, including LSTM (Long Short Term Memory), GRU (Gated Recurrent Unit), were analysed based on two text encoding schemes: word2vec and BERT (Bidirectional Encoder Representations from Transformers). The results show that some architectures are more suitable for some particular datasets, suggesting that the use of a combination of different models would offer advantages in terms of the detection performance.

Another noticeable development is the attention related method. A global-local attention network was proposed in [8]. The local semantic and global structural information are jointly encoded for better rumor detection performance. An ensemble neural architecture was presented in [15], which incorporates word attention and context from the author to enhance the classification performance.

For the graph based approach, Huang et al. [6] proposed a model based on graph convolutional networks to capture user behaviour effectively for rumor detection. In order to take the three aspects of rumor detection: contents, users, and propagation into consideration, the model is composed of three modules: a user encoder, a propagation tree encoder and an integrator that integrates the output of the two modules. A bidirectional graph convolutional network architecture was proposed in [7], in which both the upward propagation and downward propagation of information in a twitter tree were considered to enhance the rumor detection performance.

Instead of applying existing deep learning architecture to detect rumors, we will study the graph neural network for rumor detection at a lower level in this paper, and present a novel architecture based on effective and efficient aggregation layers.

## 3. SAGNN: Simplified Aggregation Graph Neural Networks

### 3.1. Preliminary: Graph Convolutional Networks

Consider a network defined by a graph $\mathcal{G} = \{\mathcal{V}, \mathcal{E}\}$. A classical graph convolutional layer proposed in [16] is given by

$$Z = \sigma(\hat{A} \cdot X \cdot W) \tag{1}$$

as shown in Figure 2, where

$$\hat{A} = \tilde{D}^{-\frac{1}{2}} \tilde{A} \tilde{D}^{-\frac{1}{2}} \tag{2}$$

with

$$\tilde{A} = A + I, \tag{3}$$

where $A \in \mathbb{R}^{|\mathcal{V}| \times |\mathcal{V}|}$ is the adjacency matrix of the graph, and

$$\tilde{D} = \begin{bmatrix} \tilde{d}_1 & & \\ & \ddots & \\ & & \tilde{d}_{|\mathcal{V}|} \end{bmatrix} \tag{4}$$

with

$$\tilde{d}_i = \sum_{j=1}^{|\mathcal{V}|} \tilde{A}_{ij} \tag{5}$$

**Figure 2.** A classical graph convolutional network (GCN) convolution layer.

### 3.2. Motivation for SAGNN

Given a source tweet $r$, and the responses or retweets $t_1$, ..., $t_{M_0}$ associated with $r$, let us call $(t_i)_{i=1}^{M_0}$ the children of $r$. Then, the set of twitters/responses containing $r$ and its children, together with the children of its children and so on, form a tree rooted at $r$, see Figure 1, with each node representing a post. Denote the set of nodes of the tree by $\mathcal{V}$.

It has been noticed that the responses to true rumors and false ones display different characteristics: when a post denies a false rumor, it tends to spark supportive or affirmative replies confirming the denial; in contrast, denial to a true rumor tends to trigger question or denial in its replies [9]. This is not surprising, since a true rumor tells truth (This might sound a bit confusing), so it is more likely to get affirmative responses. On the contrary, a false rumor communicate fabricated messages (Again, a bit confusing), and hence is more likely to get negative responses, or to be questioned by its readers. This suggests that it might be possible to distinguish true rumors from false ones by considering the interaction between a tweet and its children. In fact, based on the above observation, Ma et al. [9] proposed a recursive neural network for rumor representation learning and classification. However, the recursive architecture is not quite efficient from a computational perspective. For example, it does not easily lend itself to parallel implementation.

Recently, network representation learning has aroused a lot of research interest [17–19]. The problem of network representation is to extract features of networks, and embed the features in a low dimensional Euclidean space. Graph neural networks (GNN) turn out to be a powerful tool for this undertaking. In particular, graph convolutional network (GCN) has been widely studied for network representation. Since the tree representing a twitter and its responses is a particular case of networks, it is not surprising to find that GCNs provide an efficient solution for rumor detection as well. See [6,7] for examples of applying GCN to rumor detection. Compared to the recursive architecture proposed in [9], the GCN approach is significantly more efficient. This is an important advantage for practical applications, considering the large amount of information to be processed.

From the perspective of rumor detection, the operation of $\hat{A} \cdot X$ in Equation (1) can be thought of as calculating the interaction between a tweet and its children, as shown in Figure 3. However, the $\hat{A}$ given by Equation (2) is fixed, thus a GCN might not be able to capture the interactions in an optimal way.

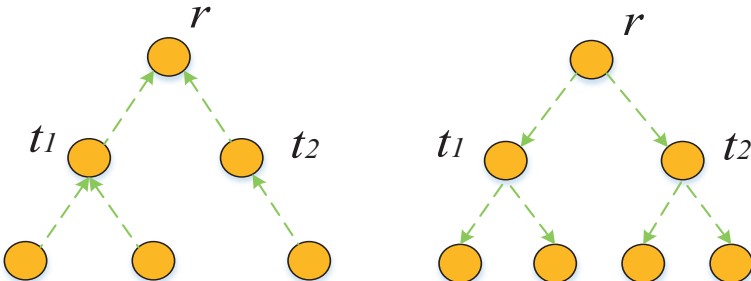

**Figure 3.** Aggregation operations performed by a GCN, in which the aggregation coefficients are fixed in the training process.

To address the issue mentioned above, we propose a simplified graph neural network architecture, using the inherent aggregation mechanism of the graph neural network to calculate the interaction between a tweet and its children. The overall architecture is shown in Figure 4.

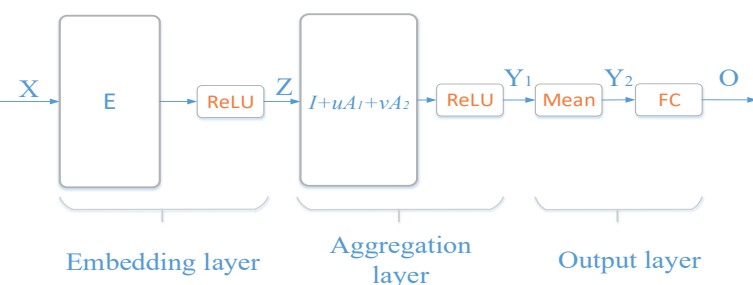

**Figure 4.** Simplified aggregation graph convolutional network.

### 3.3. SAGNN Architecture

The proposed network contains an embedding layer, one or more aggregation layer and an output layer. These layers are detailed as follows.

#### 3.3.1. Embedding Layer

Let $N_0$ be the size of the vocabulary containing all words in the considered twitters. In our experiments, $N_0 = 5000$. The task of the embedding layer is to convert the one-hot encoding of the words in the tweets into vectors of a $N$-dimensional space. The embedding layer can be represented as

$$Z = \sigma(X \cdot E), \tag{6}$$

where $E \in \mathbb{R}^{N_0 \times N}$ a matrix, $X = (x_{ij}) \in \mathbb{R}^{|\mathcal{V}| \times N_0}$ are the one-hot encoding of the tweet words, $\sigma$ is a nonlinear function, which is taken to be the *ReLU* function as usually done in neural networks. $N$ is a super-parameter, and is the dimension of output features from the embedding layer.

#### 3.3.2. Aggregation Layers

The heart of the network is the aggregation layers. The purpose of the aggregation layers is to implement learnable aggregation operations so as to capture the interactions between tweets and their children/parents in an optimal manner, as shown in Figure 5.

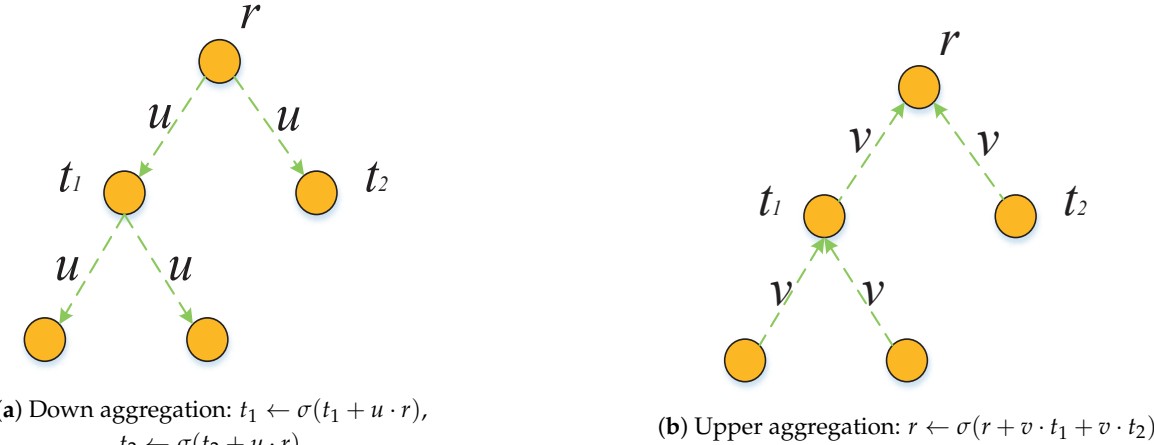

**(a)** Down aggregation: $t_1 \leftarrow \sigma(t_1 + u \cdot r)$,
$t_2 \leftarrow \sigma(t_2 + u \cdot r)$

**(b)** Upper aggregation: $r \leftarrow \sigma(r + v \cdot t_1 + v \cdot t_2)$

**Figure 5.** Learnable Aggregation Operations: The Aggregation Coefficients $u, v$ are Optimized in the Training Process.

Mathematically, the aggregation layer can be represented as

$$Y_1 = \sigma((I + u \cdot A_1 + v \cdot A_2) \cdot Z), \tag{7}$$

where $Z$ is the output of the embedding layer or the previous aggregation layer.

Compared to the classical graph convolutional layer as shown in Figure 2 [16], ourvsimplified aggregation layer does not contain the weight matrix $W$. This can also be interpreted as fixing $W$ to be an identity matrix, to put things in the framework of graph convolutional networks. Moreover, in classical GNNs, $\hat{A}$ is defined by Equation 2, while in our model, the matrix is given by

$$\check{A} = I + u \cdot A_1 + v \cdot A_2, \tag{8}$$

where $A_1 = (p_{ij}) \in \mathbb{R}^{|\mathcal{V}| \times |\mathcal{V}|}$ and $A_2 = (c_{ij}) \in \mathbb{R}^{|\mathcal{V}| \times |\mathcal{V}|}$ are the parent adjacency matrix and the children adjacency matrix, respectively. More specifically,

$$p_{ij} = \begin{cases} 1, & \text{if node } j \text{ is the parent of node } i, \\ 0, & \text{else;} \end{cases} \tag{9}$$

and

$$c_{ij} = \begin{cases} 1, & \text{if node } j \text{ is a child of node } i, \\ 0, & \text{else.} \end{cases} \tag{10}$$

While $\hat{A}$ in traditional GCN is fixed, the matrix $\check{A}$ in our SAGNN contains two learnable parameters $u$ and $v$, to distinguish the interactions from a node to its children and those from a node to its parent.

### 3.3.3. Output Layer

$$Y_2 = \text{Mean}(Y_1), \tag{11}$$
$$O = \text{FC}(Y_2), \tag{12}$$

where $Y_1 \in \mathbb{R}^{|\mathcal{V}| \times N}$ is the output of the last aggregation layer, Mean represents a mean operation over the rows of $Y_1$, and FC is a fully connected linear layer of neural network.

### 3.4. Learning Algorithm

As mentioned above, the SAGNN contains two learnable parameters $u$ and $v$ for each aggregation layer. Furthermore, the embedding layer contains a matrix $E$, and the output layer is a fully connected one, which is essentially a matrix. These constitute the learnable

parameters of the SAGNN network. To determine these parameters, an optimization procedure is applied on a properly chosen loss function.

Cross Entropy Loss Function

For classification problem, the cross entropy loss function is usually taken as the optimization objective function. In other words, we try to minimize [20]

$$L(\hat{y}, y) = -\sum_i^C y_i \log(\hat{y}_i),\tag{13}$$

where $C$ is the number of classes, $y = (y_1, \ldots, y_C)$ is the label vector with the only nonzero element being 1 at the $i$'th position if the sample is drawn from class $i$, $\hat{y}$ is the estimation given by

$$\hat{y} = \text{Softmax}(O),\tag{14}$$

where $O$ is the output of the network shown in Figure 4.

## 4. Experiments

### 4.1. Datasets

The datasets used in the experiment are two publicly available datasets Twitter15 and Twitter16 [4]. Each of the two datasets is divided into five subsets. More specifically, Twitter15 is divided into five subsets denoted by Twitter15$_0$, ..., Twitter15$_4$ respectively, with each subset further divided into a train dataset and a test dataset.Twitter16 is divided in a similar manner. The dataset contains four classes of twitters: non-rumors, false rumors, true rumors and unverified rumors.

### 4.2. Network Setup

We apply the SAGNN on datasets Twitter15 and Twitter16 to evaluate its performance. A GCNII (Graph Convolutional Network via Initial residual and Identity mapping) network [21] serves as the baseline to assess the proposed algorithm. In this experiment, the SAGNN has two aggregation layers, unlike the one shown in Figure 4, in which only a single aggregation layer is shown for clarity. Correspondingly, the GCNII has two convolutional layers.

The input feature $X$ is a $|\mathcal{V}| \times N_0$ matrix, with $|\mathcal{V}|$ being the number of tweets in the input tweet tree, $N_0 = 5000$ the size of the vocabulary. Each row of $X$ is the sum of the one-hot encoding of all words in a tweet. The output of the network is a four-dimensional vector given by Equation (12). By applying the Softmax function to the output vector, as given by Equation (14), one can get a four-dimensional probability vector, with each of its components representing the probability that the source tweet is a non-rumor, a false rumors, a true rumor and an unverified rumor, respectively.

For the loss function, a square regularization term is added for both the SAGNN network and the GCNII network. Meanwhile, dropout layers are applied to both networks to overcome possible overfitting. The stochastic gradient algorithm is adopted in the training process to find the values of the learnable parameters.

All of the calculation is performed on a laptop with an Intel i7 CPU and a Geforce 940 M GPU, using the open source machine learning framework PyTorch. The implementation of the GCNII network is based on the geometric deep learning extension library for PyTorch [22].

### 4.3. Results

The results for Twitter15 and Twitter16 are shown in Tables 1 and 2, respectively. The Acc column gives the accurary for each subset, and is equal to the number of correct predictions for the subset divided by the total number of samples in the subset. The F1 scores for each class of rumors: non-rumors, false rumors, true rumors and unverified rumors are shown in the last four columns, respectively. Examining these results, we see that the proposed SAGNN architecture and the more complicated GCNII give compara-

ble results. For the whole dataset Twitter 15, which consists of dataset T150, ..., T154, the SAGNN actually outperforms the GCNII. On the other hand, for datasets T160 and T162, the SAGNN gives lower Acc scores compared to those given by the GCNII. However, the overall performance of the SAGNN is still better than the GCNII.

Moreover, it is interesting to note the evolution of the weights $u$ and $v$ during the training process shown in Figures 6 and 7. It can been seen that these curves converge with the training iterations. The variations of these weights show the same pattern for the two distinct datasets, though having different values. This demonstrates that it is indeed quite beneficial to learn the matrix $\hat{A}$ for better rumor detection performance, instead of keeping them fixed as usually done in traditional GCNs. This might explain why the simpler SAGNN network can compete the more complicated GCNII network.

**Table 1.** Results for dataset Twitter 15. NR: Non-Rumor; FR: False Rumor; TR: True Rumor; UR: Unverified Rumor.

| Dataset | Method | Acc | F1 | | | |
|---------|--------|-----|-----|-----|-----|-----|
| | | | NR | FR | TR | UR |
| T150 | SAGNN | **0.857** | 0.851 | 0.892 | 0.867 | 0.826 |
| | GCNII | 0.823 | 0.796 | 0.85 | 0.864 | 0.786 |
| T151 | SAGNN | **0.845** | 0.844 | 0.857 | 0.895 | 0.784 |
| | GCNII | 0.813 | 0.810 | 0.829 | 0.896 | 0.725 |
| T152 | SAGNN | **0.796** | 0.846 | 0.817 | 0.810 | 0.706 |
| | GCNII | 0.773 | 0.775 | 0.790 | 0.834 | 0.698 |
| T153 | SAGNN | **0.792** | 0.75 | 0.790 | 0.907 | 0.718 |
| | GCNII | 0.768 | 0.703 | 0.763 | 0.884 | 0.723 |
| T154 | SAGNN | **0.802** | 0.8 | 0.771 | 0.824 | 0.813 |
| | GCNII | 0.769 | 0.761 | 0.719 | 0.861 | 0.742 |

**Table 2.** Results for dataset Twitter 16. NR: Non-Rumor; FR: False Rumor; TR: True Rumor; UR: Unverified Rumor.

| Dataset | Method | Acc | F1 | | | |
|---------|--------|-----|-----|-----|-----|-----|
| | | | NR | FR | TR | UR |
| T160 | SAGNN | 0.764 | 0.526 | 0.783 | 0.877 | 0.791 |
| | GCNII | **0.802** | 0.718 | 0.849 | 0.873 | 0.731 |
| T161 | SAGNN | **0.869** | 0.769 | 0.881 | 0.974 | 0.846 |
| | GCNII | 0.841 | 0.732 | 0.875 | 0.974 | 0.778 |
| T162 | SAGNN | 0.816 | 0.817 | 0.836 | 0.919 | 0.698 |
| | GCNII | **0.847** | 0.824 | 0.853 | 0.947 | 0.769 |
| T163 | SAGNN | 0.726 | 0.636 | 0.737 | 0.867 | 0.7 |
| | GCNII | **0.790** | 0.776 | 0.794 | 0.879 | 0.762 |
| T164 | SAGNN | **0.802** | 0.769 | 0.771 | 0.9 | 0.8 |
| | GCNII | 0.753 | 0.625 | 0.788 | 0.872 | 0.735 |

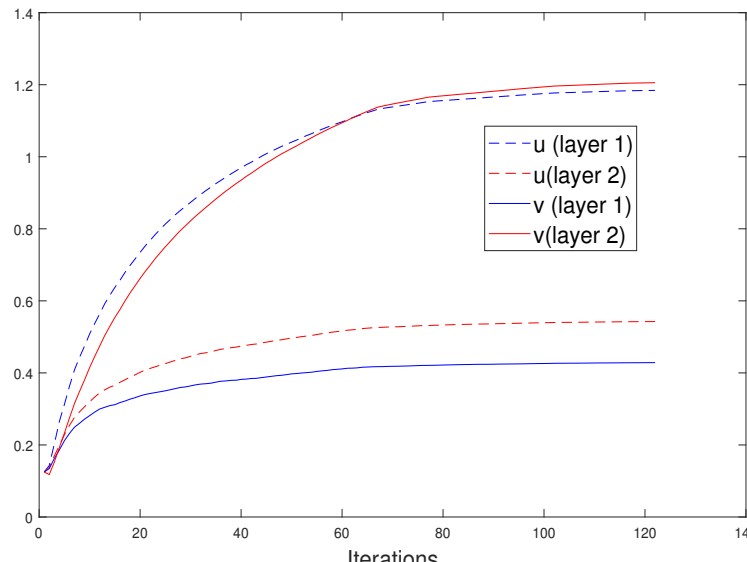

**Figure 6.** Evolution of weights *u* and *v* of the two aggregation layers during the training process for the dataset T153. Here, layers 1 and 2 refer to the first and second aggregation layer, respectively.

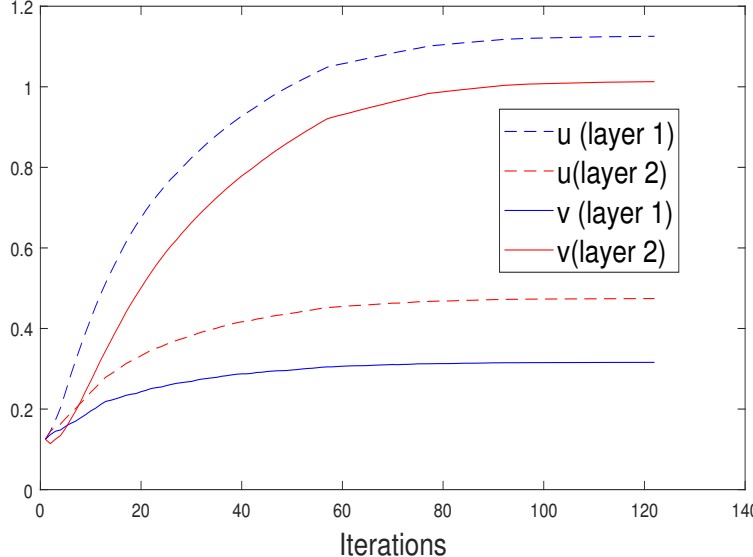

**Figure 7.** Evolution of weights *u* and *v* of the two aggregation layers during the training process for the dataset T160. Here, layer 1 and layer 2 refer to the first and second aggregation layer, respectively.

## 5. Conclusions

The observation that different types of rumours trigger different interactions between source tweets and their responses suggests that it is possible to classify rumours by aggregating the information around each tweet using graph neural networks. This motivates us to present a simplified aggregation layer to boost rumor detection performance. The neural network based on the proposed simplified aggregation layers gives comparable to or even better results than the more complicated GCNII architecture, suggesting that the learnable aggregation operation is beneficial to capture different characteristics of distinct rumours. Besides its rumor detection effectiveness, the SAGNNs has the advantage of significantly reduced computational complexity: the aggregation layer contains only two learnable parameters, in contrast to the usual convolutional layer of the conventional GCNs, in which a weight matrix $W$ of $N^2$ parameters are to be found during the training process, with $N$ being the dimension of the embedded feature vectors.

Moreover, the proposed simple aggregation layers can be further applied in more complicated architectures such as those proposed in [6,7], to implement efficient and more powerful rumor detection systems.

**Author Contributions:** Conceptualization, L.Z., J.L.; investigation: J.L., L.Z.; software, J.L., L.Z.; funding acquisition, B.Z., Y.J.; supervision, B.Z., Y.J. All authors have read and agreed to the published version of the manuscript.

**Funding:** This research was partly funded by the National Key Research and Development Program of China grant number 2018YFC0831703.

**Conflicts of Interest:** The authors declare no conflict of interest. The funder had no role in the design of the study; in the collection, analyses, or interpretation of data; in the writing of the manuscript, or in the decision to publish the results.

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
