# Peer review of "Rumor Detection Based on SAGNN: Simplified Aggregation Graph Neural Networks"

_make, doi:10.3390/make3010005_

Round 1

Reviewer 1 Report

The paper is interesting and the topic significant for the community. However, towards a possible resubmission, I have the following comments and suggestions for the authors:
- There are no clear insights over the performance of SAGNN. How do the authors tackle that issue? A solid elaboration is missing.
- I am not sure I understand how SAGNN assigns its decision value. This is not described at all, thus it is not easy for the research community to place the authors’ contribution in the related literature.
- The related work in the field is not well supported. Also, I do not find a clear sight on what is the authors’ differentiation and added value as far as their approach is concerned. This is related with the above-mentioned comments.
- The paper is not mature for publication. There are a lot of typos, while there are many cases where some abbreviations are not at all explained to the reader who struggles to understand their meanings. Tables 1 and 2 seem to be quoted hurriedly and with not proper explanation in respect to what is written in the manuscript.

Author Response

Dear reviewer,

Thanks for your valuable suggestions on improving the manuscript. The manuscript has been thoroughly revised. Here are the changes and responses we have made based on your comments.

*************************************************
Comment 1: There are no clear insights over the performance of SAGNN. How do the authors tackle that issue? A solid elaboration is missing.

The key difference between the proposed SAGNN network and the usual GCN lies that modified adjacent matrix \hat{A} is fixed in GCNs, hence it might not be able to caption the interaction between tweets in an optimal way. By contrast, the aggregation layer of SAGNN is learnable, this should facilitate to caption the interaction mentioned above, and this is beneficial for better rumor detection performance. This might explain in a intuitive way why the simple SAGNN networks can compete the more complicated GCNII network. However, further research is still necessary to better understand the dynamics of the proposed SAGNN.

**************************************************
Comment 2: I am not sure I understand how SAGNN assigns its decision value. This is not described at all, thus it is not easy for the research community to place the authors’ contribution in the related literature.

The output of the network is a four-dimensional vector given by Eq.12. By applying the softmax function to the output vector, as given by Eq. 14, we can get a four-dimensional probability vector, with each of its components representing the probability that the source twitter is an non-rumor, a false rumors, a true rumor and an unverified rumor, respectively. See lines 170-175.

**************************************************
Comment 3: The related work in the field is not well supported. Also, I do not find a clear sight on what is the authors’ differentiation and added value as far as their approach is concerned. This is related with the above-mentioned comments.

The related work section has been expanded, see page 3. Also, the motivation of the research, together with its connection to existing rumor detection work, is now more clearly described. See page 4. Also, we emphasize that this research focuses on a lower level of the graph network. That is, our contribution lies in proposing a simplified but effective aggregation layer for better rumor detection performance, instead of applying existing GCN model to rumor detection applications.

**************************************************
Comment 4: The paper is not mature for publication. There are a lot of typos, while there are many cases where some abbreviations are not at all explained to the reader who struggles to understand their meanings. Tables 1 and 2 seem to be quoted hurriedly and with not proper explanation in respect to what is written in the manuscript.

Almost all the texts in the first version of the manuscript have been rewritten. Hopefully, all typos have been removed. The abbreviations are explained so that the reader can better understand the terms.

Tables 1 and 2 have been redesigned and more details about the results have now been given, see page 8 and section 3.3.

Reviewer 2 Report

Reclamations:

1) Pictures are very poor in their quality  and the relation to the research description.  Text labels on the pictures are extremely small and hardly readable.

2) The background section is very formal and does not provide enough information about current status of relevant research. The deeper review of Deep learning methods is to be expected.

3) The definition of rumors in line 10 is ill posed in principle. Such definition contradicts to the description given in lines 75-79.

4) The main contribution, the new architecture SAGNN, is described in details, but the idea its performance is still obscured. The output value of the SAGNN is not defined at all. What is it? Probability? Binary solution? Similarity to some target value?

5) Tables 1 and 2 look like graveyard of numbers. It's impossible to make a choice between SAGNN, GCNII or even make a suggesting looking at tables. Meaning of column titles NR, FR, TR, UR is not defined.

Author Response

Dear reviewer,

Thanks for your valuable suggestions on improving the manuscript. The paper has now been thoroughly revised. Here are the changes we have made according to your comments.

1) Pictures are very poor in their quality and the relation to the research description. Text labels on the pictures are extremely small and hardly readable.

The pictures have now been redrawn and each caption has been rewritten to give a more clear description of the figure. The size of the texts in the pictures have been adjusted to improve the readability. Also, the figures are now more fully described in the revision. But it is hard to list all the changed lines and pages here, because the changes are scattered around the whole paper. Please see the revised paper and the diff file.

2) The background section is very formal and does not provide enough information about current status of relevant research. The deeper review of Deep learning methods is to be expected.

The first two sections on the background and related work have been expanded. Also, the deep learning method has been more fully and systematically accounted, including convolutional network based methods, attention based methods and graph network based methods. see the revised first two sections on pages 1, 2 and 3

3) The definition of rumors in line 10 is ill posed in principle. Such definition contradicts to the description given in lines 75-79.

The definition of rumors has now been reformulated, and the reference is now cited so that the interested reader can get further historical and philosophical information about rumor study.

The definition of rumors in line 10 is in fact in agreement with the description of the lines 75-79 in the first version of the manuscritp. More specifically, a true rumor tells truth (This might sound a bit confusing), so it is more likely to get affirmative responses. On the contrary, a false rumor communicate fabricated messages (Again, a bit confusing), and hence is more likely to get negative responses, or to be questioned by its readers. Therefore, if somebody denies a true rumor, his denial tends to trigger question or denial in its replies. Similarly, a denial to the false rumor tends to spark supportive or affirmative replies confirming the denial.

These lines have been reformulated in order to avoid confusion. See lines 106-113.

4) The main contribution, the new architecture SAGNN, is described in details, but the idea its performance is still obscured. The output value of the SAGNN is not defined at all. What is it? Probability? Binary solution? Similarity to some target value?

The motivation of SAGNN is now more fully accounted, see section 2.2 on page 4.

The output of the network is a four-dimensional vector given by Eq.12. By applying the softmax function to the output vector, as given by Eq. 14, we can get a four-dimensional probability vector, with each of its components representing the probability that the source twitter is an non-rumor, a false rumors, a true rumor and an unverified rumor, respectively. See lines

The key difference between the proposed SAGNN network and the usual GCN lies that modified adjacent matrix \hat{A} is fixed in GCNs, hence it might not be able to caption the interaction between tweets in an optimal way. By contrast, the aggregation layer of SAGNN is learnable, this should facilitate to caption the interaction mentioned above, and this is beneficial for better rumor detection performance. This might explain in a intuitive way why the simple SAGNN networks can compete the more complicated GCNII network. However, further research is still necessary to better understand the dynamics of the proposed SAGNN.

5) Tables 1 and 2 look like graveyard of numbers. It's impossible to make a choice between SAGNN, GCNII or even make a suggesting looking at tables. Meaning of column titles NR, FR, TR, UR is not defined.

The tables have now been redesigned. Hopefully, the readability has been improved. Also, NR, FR, TR, UR is now defined, see the two tables on page 8.

Round 2

Reviewer 1 Report

The authors did a reasonably effort to improve their submission. My comments and concerns were answered in a large extend. I feel that the paper can be published in its current form.